# The Roles of Vitamin D and Polyphenols in the Management of Age-Related Macular Degeneration: A Narrative Review

Nádia Fernandes [1,2], Marta Castro Araújo [2] and Carla Lança [1,3,*]

1   Instituto Politécnico de Lisboa, Escola Superior de Tecnologia da Saúde de Lisboa (ESTeSL), 1990-096 Lisbon, Portugal; nadia.fernandes@estesl.ipl.pt
2   Topcare Clinic, 2780-117 Oeiras, Portugal
3   Comprehensive Health Research Center (CHRC), Escola Nacional de Saúde Pública, Universidade Nova de Lisboa, 1600-560 Lisboa, Portugal
*   Correspondence: carla.costa@estesl.ipl.pt

**Abstract:** Age-related macular degeneration (AMD) is a chronic progressive ocular disease and the main cause of severe visual impairment in the elderly. Vitamin D deficiency may be a risk factor for AMD. Additionally, current evidence suggests dietary advice of increasing consumption of polyphenols, which may have antioxidant and anti-inflammatory properties. The aim of this review was to describe the roles of vitamin D levels and polyphenols in the management of AMD. The results of this review showed mixed evidence regarding the protective effect of vitamin D against AMD. Polyphenols (flavonoids group, curcumin and resveratrol) seem to play an important role as angiogenesis inhibitors, but their effect on AMD is still unclear. Vitamin D and polyphenols may both play an important role as nutritional modifiable protective factors that reduce the risk of AMD progression. However, more research is necessary to better understand the roles of vitamin D and polyphenols in different stages of the disease.

**Keywords:** age macular degeneration; vitamin D; 25-hydroxyvitamin D serum; polyphenols; nutrition; eye healthcare; retina; choroid





## 1. Introduction

Age-related macular degeneration (AMD) is a chronic, progressive ocular disease that causes central vision loss. In Europe, AMD represents the main cause of severe visual impairment and blindness in the elderly. Approximately 67 million people worldwide are currently affected by this disease, and an increase by 15% in its prevalence is expected by 2050 in the European population [1]. AMD is responsible for 8.7% of blindness throughout the world, and it is the main cause of visual impairment in developed countries in individuals aged 60 years or more [2]. The incidence of AMD is variable. The incidence of late AMD is higher in Europe, and Africa has the highest incidence of early AMD. Annual incidence of late AMD in American whites was 0.35% or 0.14% in Europeans [3].

AMD is a multifactorial disease affected by non-modifiable (age, family history, genetics, and ethnicity) and modifiable risk factors (health status, smoking habits, low antioxidant diet, sedentary lifestyles, and exposure to ultraviolet [UV; 240–300] and blue light [415–455]). The prevalence of AMD is increasing due to the growth of the aging population. Thus, health promotion and preventive medicine are essential tools to reduce the risk of visual impairment [4].

Oxidative stress has an important role in the pathogenesis of AMD, but the exact causal mechanism is not clear yet [5]. The overproduction of reactive oxygen species (ROS) and the appearance of free radical-mediated oxidative damage with consequent hypoxia, leads to a chronic inflammatory process [5]. Retinal regeneration decreases with age [6]. Additionally, ROS increase with an improper diet.

Antioxidants, vitamins, and minerals have been compounded into dietary supplements to aid in the prevention and to slow progression of some forms of AMD [6]. Some nutritive supplements with antioxidants properties are based on the Age-Related Eye Disease Study formulation (AREDS 1 and AREDS 2; e.g., Nutrof®, Remiren®, Meralut®, and Vitol®). These supplements also contain vitamin D (Vit D) and polyphenols (blueberry and resveratrol), contributing to improved retinal antioxidant defense. Vitamin D and dietary polyphenols protect against the damage caused by free radicals [5]. Antioxidants, such as vitamins C and E, carotenoids (lutein and zeaxanthin), polyphenols, zinc, and copper, together with a Mediterranean diet and/or dietary supplements, have been shown to reduce the risk of AMD progression to late stages [6]. Carotenoids are the main components of the macular pigment, and their protective role relies on antioxidant properties [6].

Vit D, also known as 25-hydroxyvitamin D [25(OH)D], is a steroid hormone that is synthesized in the skin from cholesterol, in a reaction catalyzed by sunlight. 25(OH)D influences the immune system, and its deficiency has been associated with various chronic diseases, such as autoimmune and cardiovascular diseases, osteoporosis, diabetes, and cancer [7]. Therefore, it is considered one of the most relevant health issues [8].

Vit D deficiency may be a risk factor for several eye diseases, including AMD [9]. Vit D receptors have been shown to be present in retinal vascular and choroidal endothelial cells [9]. Previous reviews have reported the anti-inflammatory potential of Vit D in neovascular AMD and its effect on the reduction of the risk of early AMD development [6]. The mechanism of action seems to protect against oxidative damage in the cellular membrane and proteins [10].

Current evidence suggests that patients with AMD should be given advice to increase their consumption of flavonoids, which may have antioxidant and anti-inflammatory properties [11]. The role of polyphenols has been studied, demonstrating a preventive effect against chronic oxidative stress and a reduction in anti-vascular endothelial growth factor (VEGF) side effects. Polyphenols have a preventive effect against chronic oxidative stress: reducing VEGF side effects. Resveratrol and its analogs are promising agents that can improve the anti-VEGF therapy in exudative AMD [12]. Furthermore, polyphenols seem to be promising in reversing oxidative stress and inflammation-associated damage. There is some evidence that some polyphenolic groups affect vascular health through improved endothelial function and vascular function, thereby possibly improving the management of AMD [12]. However, there is limited research in this field of study [11].

In this narrative review, we searched PubMed and Web of Science databases to investigate the effects of Vit D and polyphenols on AMD progression. Vit D and polyphenols have anti-oxidative and anti-inflammatory properties which impact AMD pathogenesis. Vit D3 (cholecalciferol), as well as some polyphenols, such as, flavonoids, resveratrol, and curcumin, are included in some nutritional supplements together with the AREDS formulation. We manually searched the references of evidence-based reviews/meta-analyses and practice guidelines, and selected randomized controlled trials (RCTs) and observational studies (case-control, cohort prospective, and cross-sectional studies) published between 2017 and 2022. Preclinical studies and studies included in previous reviews/meta-analysis were excluded. The aim of this study was to describe the roles of Vit D and polyphenols in the management of AMD.

## 2. AMD and Vit D

### 2.1. Clinical Diagnoses

There are two forms of AMD, dry and wet. AMD is also classified by three stages: early, intermediate, or advanced (neovascular macular degeneration and geographic atrophy (GA)). Wet AMD is treated with inhibitors of vascular endothelial growth factor (anti-VEGF), and there is no treatment for GA, the more severe form of dry AMD [6]. Structural changes in the progression of AMD can be analyzed through ocular fundus retinography (Figure 1A) and spectral domain optical coherence tomography (SD-OCT; Figure 1B) as routine structural ophthalmic images analysis for qualitative and quantitative assessment

of the retina and choroid [13]. SD-OCT allows to study retinal layers cross-sectionally, and it is a tool for the diagnosis and management of AMD, helping to monitor disease progression. Along with retinal fundus images, OCT allows retina specialists to classify AMD stages and predict AMD progression [14]. Angio OCT (OCT-A) is a non-invasive imaging technique that is applied in several retinal vascular diseases. It offers volumetric information giving structural information about the retina and choroid blood flow [14].

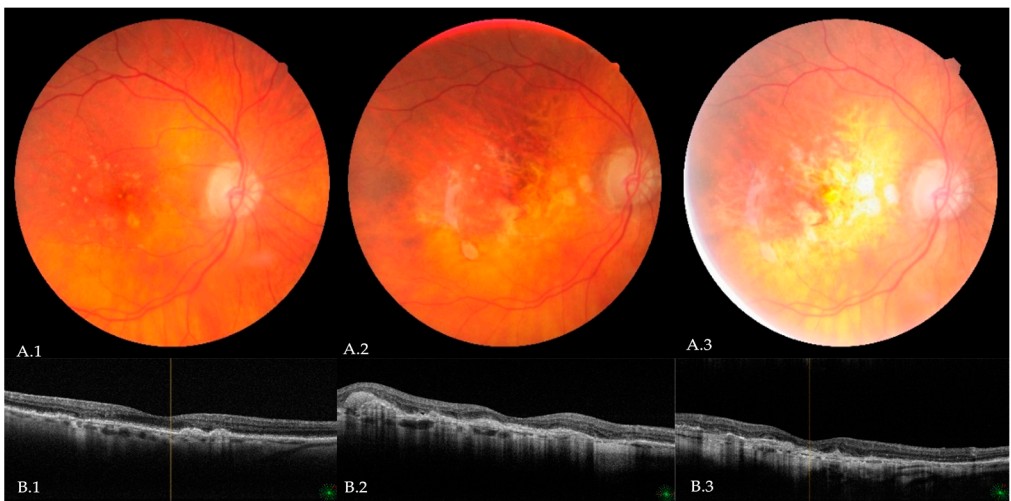

**Figure 1.** Ocular fundus retinography (**A.1**–**A.3**) and spectral domain—optical coherence tomography (**B.1**–**B.3**) showing the progression from intermediate to advanced AMD; from left to right: (**A.1**,**B.1**) large, confluent, soft drusen in high risk intermediate AMD, retinal pigment epithelium (RPE) migration and pericentric geographic atrophy; (**A.2**,**B.2**) occult choroidal neovascularization, pigmentary clumping, and RPE serous detachment with a tear in an advanced stage; (**A.3**,**B.3**) end-stage AMD with disciform scar (Araujo, M. e Fernandes, N. courtesy).

According to the AREDS classification, AMD can be divided into four categories based on ocular fundus examination and the existence of drusen, atrophy, and neovascularization: Category 1—no AMD if 0–5 small drusen are present. Category 2—early AMD if multiple small drusen or few intermediate-sized drusen are present, or macular pigmentary change. Category 3—intermediate AMD characterized by extensive intermediate-sized drusen, or at least one large drusen, or GA not involving the macular center. Category 4—advanced AMD determined by GA involving the macular center or any sign of choroidal neovascularization (CNV) with vision loss [15].

### 2.2. Effects of Vitamin D Levels and Retinal Choroidal Function and Structure

Vit D serum 25(OH)D is an important human health indicator. Blood plasma concentrations <20 ng/mL are frequent and affect about half of the world's population. Concentrations >30 ng/mL are necessary for optimal health and solar UVB exposure. Vit D supplementation and food fortification are efficient ways to reach these concentrations. Considering the significant health benefits of 25(OH)D and the low probability of side effects, it seems important to determine the effect thresholds concentrations for different diseases [16].

Deficiency in Vit D levels is common in older people. Although Vit D may be taken in the form of nutritional supplements or Vit D food fortification (e.g., cow's milk, margarine, orange, plant-based milk, cereals, and bread) [17]. Vit D is primarily produced in the skin, and its production may be impaired in the elderly due to skin atrophy [10]. Decreased sun exposure, obesity, nutritional deficiency, clothing style, malabsorption, and liver diseases are other causes of Vit D deficiency [10]. Some health conditions cause malabsorption of Vit D, such as cystic fibrosis, inflammatory bowel disease, gastric bypass surgery, and

intestinal lymphangiectasia. Therefore, the risk for Vit D deficiency increases in those individuals [18].

The association between Vit D plasma levels and AMD has been investigated in the last decades, and some observational studies have shown an inverse relationship between Vit D plasma levels and early AMD or late atrophic and/or neovascular AMD [7]. Previous studies have shown that 25(OH)D serum deficiency may be associated with the risk of AMD, including neovascular AMD. Additionally, high levels of 25(OH)D can reduce the risk of AMD. However, results of meta-analyses do not indicate that high serum Vit D levels have a significant protective effect against the progression of any stages or different AMD forms. Thus, there is no clear evidence of a definitive association between serum 25(OH)D level and AMD [19].

In the Icel et al. case-control study (Table 1), macular microvasculature in 25(OH)D deficiency was investigated using Angio OCT (OCT-A) [20]. A serum level of <20 ng/mL was defined as 25(OH)D deficiency. The statistically significantly higher retinal micro vascularity in Group II compared to Group I may imply that the cut-off value ($\geq$20 ng/mL) is effective in terms of detecting changes in retinal health. The statistically significant difference in OCT-A examinations seems to support that this cut-off value is effective, as retinal vascular density values were lower in the group with lower 25(OH)D levels. Additionally, in this group, inversely proportional values of the fovea avascular zone were found. The authors suggested future studies with a follow-up design to better explain these findings [20]. In the case-control study of Kabataş et al., the authors found that serum Vit D levels were significantly lower in patients with AMD and in patients with advanced-stage AMD (11.4 $\pm$ 5.1 ng/mL in wet-type) compared to patients with early- and intermediate-stage AMD (15.3 $\pm$ 10.9 ng/mL) [10]. In the cross-sectional study of Serena et al., the aim was to determine Vit D levels in different AMD stages. In their study, plasma 25(OH)D concentrations were defined in deficiency categories as severe (<10 ng/mL), deficiency (10–19 ng/mL), insufficiency (20–29 ng/mL), and sufficiency ($\geq$30) [21]. The study results revealed that Vit D levels were decreased in patients affected by AMD compared to controls. Nevertheless, the authors concluded that it is difficult to determine the precise correlation between Vit D deficiency and AMD progression and emphasize the different 25(OH)D concentration blood levels used by the different studies [21].

**Table 1.** Association between vitamin D levels and AMD.

| Study Design, Country | Vitamin D (Mean Serum 25(OH)D Levels ng/mL) | N (Group I e II) | Age (Mean $\pm$ SD) | Association between Vit D and Retinal-Choroidal Structure/AMD | Author(s) Study, Year |
|---|---|---|---|---|---|
| Case-Control | Group I: deficiency = 7.61 | 82 | 37.29 $\pm$ 12.76 | Deficient Vit D levels affected macular perfusion with lower retinal vascular density values. Central macular volume and RNFL were not significantly different. | Icel, et al., 2022 [20] |
| | Group II: control = 25.29 | 50 | 39.1 $\pm$ 11.59 | | |
| Case-Control | AMD group = 14.4 $\pm$ 9.6; (wet-type AMD = 11.4 $\pm$ 5.1 dry-type AMD = 15.3 $\pm$ 10.9) | 114 (64/50) | 71.5 $\pm$ 7.9 | Significant correlation between vit D deficiency and AMD progression. | Kabataş et al., 2022 [10] |
| | Control group = 29.4 $\pm$ 14.6 | 102 (57/45) | 69.4 $\pm$ 10.1 | | |
| Cross-sectional | AMD group = 15 $\pm$ 10 | 93 (57/36) | | Deficient Vit D levels (<30 ng/mL) were found in 89.2% of the AMD group and 50.5% had higher prevalence of Vit D deficiency comparatively with controls 31.2%. | Serena, et al., 2022 [21] |
| | early AMD = 12.5 $\pm$ 7.3 | (10) | | | |
| | intermediate AMD = 15 $\pm$ 11 | (12) | 78.96 $\pm$ 8.5 | | |
| | advanced aAMD = 15 $\pm$ 8 | (19) | | | |
| | advanced nAMD = 17 $\pm$ 11.5 | (52) | | | |
| | Control group = 21 $\pm$ 14 | 93 (54/39) | 78.8 $\pm$ 8.4 | | |

OCT: optical coherence tomography; Vit D: vitamin D; 25(OH)D: 25-hydroxyvitamin D; CT: choroidal thickness; AMD—age macular degeneration; RNFL—retinal nerve fiber layer thickness; SD: standard deviation.

### 2.3. Vitamin D and Polyphenols Supplementation and Retinal Choroidal Function and Structure

A prospective study conducted by Öncül et al. (Table 2) demonstrated that mean Vit D values were $10.94 \pm 3.88$ ng/mL in intervention group I and $27.67 \pm 7.26$ ng/mL in control group II [9]. Choroidal thickness (CT) in group I was significantly lower than in group II. After Vit D supplementation, CT increased significantly compared with baseline measurements, but there were no significant differences in central macular thickness values post-treatment versus baseline. The study was limited by the age of the participants, as only individuals between the age of 18 and 40 years were included.

Polyphenols or phenolic compounds are a group of more than 8000 diverse phytochemicals with a phenolic structure. They are found in plants and beverages, such as vegetables, fruits, chocolate, tea, and wine. Polyphenols are classified according to their chemical structure and can be divided into groups: flavonoids (flavanols, flavan-3-ols, flavones, flavanones, and anthocyanins), phenolic acids, phenolic amides, and other polyphenols (curcumin, resveratrol). Previous research demonstrated that dietary polyphenols have a beneficial role in the prevention of chronic diseases and may improve human health through their antioxidant and anti-inflammatory activities [22].

In the pilot study conducted by Majeed et al., the efficacy and safety of Macumax® was evaluated (Table 2) [23]. This phyto-mineral nutritional supplement contains bilberry extract. Bilberry (*Vaccinium myrtillus* L.) is a kind of berry rich in anthocyanins that has many health benefits. As a group of polyphenols, anthocyanins are involved in the antioxidative process because of their potential to inhibit ROS production at the cellular level [24]. The results of the study by Majeed et al. demonstrated an improvement in functional vision, significative improvements in distant vision acuity, and a reduction in distortion in Amsler's Grid test. The improvement in central vision went from 25% to 60% from day 60 to day 90, respectively, after the treatment. These findings showed a positive effect of this supplement in early-stage AMD progression, and no side effects were registered. However, the changes in retinal structure during follow-up study were not significant [23].

**Table 2.** Vitamin D and polyphenol supplementation and dietary intake's protective effects against AMD.

| Compounds | Study Design, Country | Doses or Natural Source/Follow-Up | Age in Years (Mean ± SD) | Summary of Findings | AMD Type | Author(s), Year |
|---|---|---|---|---|---|---|
| Vit D3 (cholecalciferol) | Prospective Switzerland | 300.000 IU/month Follow-up: 3 months | $28.4 \pm 6.74$—group I (Vit D deficiency) $30.2 \pm 6.25$—group II (Normal Vit D levels) | CT values measured in OCT increased significantly after vit D supplementation. | Absence of AMD | Öncül et al., 2020 [9] |
| Macumax® (bilberry, saffron extract) | RCT India | Bilberry extract: 40 mg/day Follow-up: 3 months | $58.97 \pm 7.5$ | Significant improvement in functional vision in early stages of dry AMD with no structural AMD progression. Significant decrease of vision distortion and increased distance vision and dark adaptation. No structural changes were observed in OCT. | Dry Early-stage | Majeed et al., 2021 [23] |
| Dietary Flavonoids Intake | Cohort Study Australia | Intake of total flavonoids (median) 287.59 mg/day Follow-up: 12 months | $78.7 \pm 9.1$ | Higher intake of flavanols (quercetin) and flavan-3-ols (epigallocatechin-3-gallate and epigallocatechin) contributed to better treatment outcomes with anti-VEGF therapy in neovascular AMD. Significantly better BCVA associated with lower IRF in OCT (flavanols and flavan-3-ols). | Neovascular (anti-VEGF therapy) | Detaram et al., 2021 [25] |

RCT: randomized clinical trial; CT: choroidal thickness; OCT: optical coherence tomography; Vit D: vitamin D; AMD—age macular degeneration; IU—international unit; BCVA—better corrected visual acuity; IRF—intra-retinal fluid.

Additionally, some flavanols revealed an antiangiogenic potential that can be relevant in advanced stages of AMD. Quercetin and other dietary polyphenols may have an important role in the management of neovascular AMD. A cohort study conducted by Detaram et al. assessed if dietary intake of flavonoids was associated with an anti-VEGF outcome

in patients with neovascular AMD. The study's results demonstrated that a higher intake of food sources with quercetin, epigallocatechin-3-gallate (EGCG), epigallocatechin, and tea (tannic acid) contributed to better visual acuity and a lower risk of develop intraretinal fluid (IRF) after the anti-VEGF treatment. These results could be associated with the antiangiogenic properties of the flavonoids and their positive effect on vascular health function in retinal diseases [5]. However, a higher intake of anthocyanins was associated with a higher risk of IRF [25].

Gopinath et al. conducted a cohort study and recruited 2856 Australian adults at baseline with 2037 adults followed up for 15 years. Dietary intake was assessed by using a semiquantitative food-frequency questionnaire (FFQ). Estimates of the flavonoid content of foods in the FFQ were assessed by using the United States Department of Agriculture (USDA) National Nutrient Database for Standard Reference of Flavonoid, Isoflavone, and Proanthocyanidin databases. The results of the study showed that the median dietary intake of total flavonoids was 875 mg/d. In this study, flavonoids were associated with reduced prevalence of different AMD stages, mainly quercetin (a flavonol) and hesperidin (flavanone). The study findings suggest an independent and protective role of flavonoids, along with a reduction of the risk of developing AMD. However, there were no significant associations between the incidence of AMD and flavonoid intake after the 15 years of follow-up [11].

There are two clinical trials that are ongoing: one trial focusing on curcumin supplement's effect in AMD (clinicalTrials.gov: NCT04590196) [26] and another that aims to evaluate the safety and efficacy of a combination of resveratrol, quercetin, and curcumin (RQC) versus curcumin (C) alone in patients with AMD (clinicalTrials.gov: NCT05062486) [27]. Resveratrol is a phytochemical from the polyphenolic group that could be found in grapes, red wine, grape juice, peanuts, cocoa, and berries of *Vaccinum* species, including blueberries, bilberries, and cranberries, which is also riche in anthocyanins. Resveratrol is extracted from roots of *Polygonum cuspidatum* and from *Vitis vinifera* and may contain between less than 1 milligram (mg) to 500 mg per tablet or capsule. Its antioxidant and anti-inflammatory properties could be associated with a potential role in the prevention and treatment of AMD [28]. Resveratrol may reduce neovascularization, thereby protecting against VEGF production [29]. The combination action of anti-VEGF therapeutic and resveratrol demonstrated a reduction in the side effects treatment through increasing the phagocytosis process induced by resveratrol, which contributed to a reduction in profibrotic changes in retinal pigmentary epithelium (RPE) cells after anti-VEGF drug exposure [12]. An RCT conducted by García-Layana et al. in patients with unilateral wet AMD demonstrated a significant reduction effect on some proinflammatory cytokines (IL-6, IL-8, and TNF-$\alpha$). However, no significant effect on better corrected visual acuity (BCVA) was found with the intake of 30 mg/day of resveratrol in Retilut® for a period of 6–12 months [30].

## 3. Discussion

The purpose of this narrative review was to summarize all the available relevant evidence on the relationship between Vit D, polyphenols, and AMD (Figure 2). The results of this review showed mixed evidence regarding the protective effects of Vit D and AMD. For polyphenols and AMD, there is a scarcity of studies, and some RTCs are still ongoing.

Vit D may protect the RPE and choroidal cell function—layers that are directly related to AMD development. Thus, increasing 25(OH)D levels through dietary intake such as Vit D-enriched foods and supplementation may be beneficial [7]. Evidence seems to support 25(OH)D concentrations above 30 ng/mL (75 nmol/L) for cardiovascular disease and all-cause mortality rate, whereas the thresholds for several other outcomes appear to range between 40 and 50 ng/mL [16].

Kabataş et al. suggested that Vit D protects against the development of AMD. Cellular death and retinal aging may be improved by controlled release of diverse cytokines that decrease inflammation. Lower Vit D serum levels were found in AMD patients with choroidal neovascularization, which demonstrated that 25(OH) D3 deficiency may con-

tribute to the incidence of choroidal neovascularization-related diseases, increasing the risk of neovascular AMD. No association was found in early or GA AMD stages [10]. Vit D has antioxidant properties and protects DNA against oxidative damage, and plays an important role in the maintenance of healthy RPE functions by causing autophagy in the damaged RPE. Vit D is also able to retain antiapoptotic and antiangiogenic properties that may be important in the prevention of retinal function degeneration [10].

**Figure 2.** Vitamin D and polyphenols: molecular structure and food sources.

The results of the systematic review and meta-analysis conducted by Ferreira et al. do not confirm the possible inverse association between Vit D levels and AMD. However, there was a trend for advanced AMD in people with serum Vit D < 50 nmol/L [31]. It is logical to expect differences between selected studies from systematic reviews and meta-analyses in terms of heterogeneity, study design, and procedures, including population representativeness, ethnicity, and methods used to measure Vit D levels and AMD's presence [7]. The absence of longitudinal studies could be another reason for the lack of strong evidence [19], and dietary levels of some food sources could contribute to the unclear evidence of a definitive association between Vit D serum levels and AMD [19].

A structure and function evaluation seems to be an interesting tool for assessing changes in the retina and choroid after supplementation. The relation between serum 25(OH)D level after Vit D supplementation and CT was analyzed by Öncül et al. [9]. After Vit D supplementation, CT values increased. There was significant choroidal thinning in perifoveal areas of AMD participants compared to healthy individuals. Vit D may have a direct impact on the vascular wall due to the presence of 1a-hydroxylase, an enzyme responsible for the conversion of 25(OH)D into calcitriol, in endothelial and vascular smooth muscle cells. Other factors, such as age, blood pressure, refractive error, intraocular pressure, and axial length, could also affect choroidal structure, and it is important to take those factors in account in future research [9]. A mean 9.4-year follow-up study conducted by Merle et al. demonstrated that a high dietary intake of Vit D was associated with a 40% reduction in the risk of progression to advanced AMD [32]. The RCT study conducted by Christen et al. demonstrated that Vit D3 supplementation had no significant overall effect on incidence or progression in advanced AMD. However, there was a reduction in early and intermediate AMD stages' prevalence, and there were fewer cases of AMD progression, which may support the benefit of Vit D3 in reducing AMD's progression [33]. Several studies tested supplementation with formulations that contained various compounds [33]. This may have been a confounding factor that impaired the conclusions regarding the effects attributed to specific compounds [34]. Simultaneous administration of different compounds influences intestinal absorption and may influence AMD prevention and progression; therefore, it may be relevant to conduct research studies with isolated compounds [34]. It is also relevant to investigate if Vit D, dietary polyphenols compounds, and another micronutrients and vitamins have a synergistic effect that contributes to improvements in AMD management.

25(OH)D3 supplementation may be helpful in the prevention and management of macular degenerative diseases. Regulation of modifiable lifestyle risk factors that influence

levels of Vit D, such as nutrition support, a healthy lifestyle, and sun exposure, may impact the prevention and management of neovascular AMD [20]. It is important to note that the reduced ultraviolet B (UVB) (290–315 nm) at relatively high latitudes, particularly during the winter, may decrease cutaneous absorption and synthesis of Vit D3. Other individual factors, such as dark skin pigmentation, a sun-avoidant lifestyle, conservative clothing habits, and liberal use of sunscreen also limit cutaneous exposure [21]. Therefore, these modifiable factors can influence the levels of Vit D of participants included in these studies.

In clinical practice, patient follow-up is based on BCVA, ocular fundus images, SD-OCT, OCT-A, and fluorescein angiography. Nevertheless, according to a systematic review conducted by Csader et al., other measurements may be helpful, such as a multifocal electroretinogram (mfERG) for retinal function, to detect early changes in macular function associated with AMD [34]. The RCT study conducted by Parravano et al. assessed retinal functional changes by mfERG recordings and macular structure by SD-OCT evaluation, after 6 months of supplementation with Macuprev®, in intermediate AMD [13]. The results of the trial showed increased average responses in amplitude density in ring 1 (concentric annular retinal region centered on the fovea limited area of 0–2.5°) and ring 2 (concentric annular retinal region centered on the fovea limited area of 2.5–5°) [35], suggesting a functional improvement in the pre-ganglionic retinal elements located in the 0–5 central retinal degrees [13]. However, at the end of the follow-up, there were no differences in the macular chorioretinal structural parameters of the intervention group [13]. As a result, the improved function was not significantly correlated with the structural changes for the same retinal areas [13]. Studies with longer follow-up are necessary to confirm these results, along with the role of each compound in improving macular function [13]. The functional vision loss of AMD has a high impact on quality of life, as it affects reading and computer use, driving, and face recognition, and may be associated with depression [31]. Supplementation of AMD patients may contribute to better functional outcomes. Futures studies may include contrast sensitivity measures, as a visual function outcome, to assess if there is a qualitative effect on patient's functional vision performance in different stages of the disease.

There are more than 8000 diverse polyphenols. However, only a few have been studied in RCTs and observational studies that included patients with AMD. AMD is an age-related disease associated with chronic oxidative stress. Thus, the antioxidative, anti-inflammatory, and antiaging properties of polyphenols may play important roles in the prevention and management of AMD [12]. Natural antioxidants may act by slowing the progression of AMD, contributing to the prevention of irreversible vision loss [36]. Flavonoids are the predominant dietary polyphenols. Flavonoids act by decreasing oxidative stress and inflammatory processes, and by producing angiogenesis inhibitors. A regular diet rich in these compounds can have a relevant role in the prevention of AMD, and in inhibiting the progression of AMD [37]. Epigallocatechin gallate, found in green tea, inhibits ROS, angiogenesis, VEGF, apoptosis of retinal ganglion cells, and protects against mitochondrial damage. Quercetin, found in fruits and vegetables, inhibits ROS, VEGF, pro-inflammatory molecules, and apoptosis of the neurons, protecting RPE cells in age-related diseases [38]. Quercetin may also play an important role in AMD progression due to its capacity to reduce oxidative damage, which induces inflammation, and to reverse photoreceptor apoptosis and retinal degeneration [36].

There is limited research on promising phytochemicals with antioxidant and anti-inflammatory properties. The expression of angiogenic factors VEGF and hypoxia-inducible factor 1$\alpha$ (HIF-1$\alpha$) could be reduced by polyphenols. Flavonoids appear to be important for angiogenesis inhibitors, but the effect is unclear [11]. Other groups of polyphenols may also have an important role in AMD progression. Curcumin (*Curcuma longa*) is a polyphenolic compound with an antioxidant property which could be combined with another phenolic, resveratrol. Curcumin can inhibit lipid peroxidation, ROS, and VEGF. It also reduces the levels of pro-inflammatory cytokines and DNA damage through increasing antioxidant enzymes. Resveratrol contributes to inhibit oxidative stress, VEGF, and lipid

peroxidation; decreases inflammatory processes; and increases glutathione production [38]. A retrospective case-control study conducted by Allegrini et al. demonstrated that a curcumin supplement with AREDS2 compounds, astaxanthin and resveratrol, combined with an anti-VEGF injection, improved functional outcomes, including a reduction in injection therapy [39]. Nevertheless, additional evidence is necessary to evaluate the safety and efficacy of curcumin in AMD [36]. The therapeutic potential of resveratrol could be related to a reduction in anti-VEGF injections' side effects. Future research to investigate the synergetic effects of resveratrol and anti-VEGF in neovascular AMD may be relevant to reducing the complications of this treatment [36]. In a meta-analysis conducted by Csader et al., although curcumin was associated with improvements in visual acuity, no significant changes in central macular thickness, and a reduction in the number of anti-VFGF injections, definite conclusions about the efficacy of curcumin could not be made [34]. Future well-performed clinical studies may give detailed insights into the effects of these dietary polyphenols in AMD management and prevention. There are two clinical trials ongoing—one focusing on curcumin's effect; and another on curcumin, resveratrol, and quercetin's effect. Those studies may give more detailed insights on these polyphenols' effect on AMD progression.

Anthocyanins are present in blueberries, raspberries, blackberries, strawberries, or red grapes. One flavonoid reduces the inflammation process in chronic conditions—namely, cyanidin-3-glucoside (C3G) that is important in the prevention of retinal diseases [37]. Anthocyanins and their protective role could be linked to antioxidative and anti-inflammatory properties related to ROS generated induced by light exposure [12]. However, a higher intake of anthocyanins was associated with a higher risk of intraretinal fluid in patients with neovascular AMD [25]. More studies are necessary to understand the specific roles of anthocyanins in AMD prevention and treatment [40].

Saffron (Crocus sativus) is a phytochemical with 150 compounds. Carotenoids, crocin, and crocetin are the most organically active elements. Open-label longitudinal studies and double-blind RCTs demonstrated that saffron reduces progression of dry, mild-to-moderate, and advanced stages of AMD [36]. Long-term consumption of polyphenolic compounds may protect against some chronic disorders, such as cancers, cardiovascular, and neurodegenerative diseases [41]. The efficacy of polyphenols, bioavailability, absorptive food, and drug interactions, in terms of short- and long-term health effects and possibly health-promoting mechanisms, should be investigated in future research [41].

With aging, the composition of the microbiome changes, and several degenerative diseases are associated with microbiome aging. AMD is a multifactorial disease related to age, environmental, epigenetic, and genetic risk factors. Diet seems to have a deep impact on the onset and progression of AMD through anti-oxidative and anti-inflammatory mechanisms [42]. There is a potential interaction between the immune system modulation, inflammation, and diet on the gut microbiota. Thus, gut microbes and their metabolites may play a key role in AMD [43]. Polyphenols, such as blueberry and green tea extract drinks, have potential interactions with gut microbiota, as their metabolites may promote beneficial gut bacteria and inhibiting intrusive species [41].

Polyphenols seem to be one of the most important compounds inhibiting effects on AMD progression, through dietary intake of fruits and vegetables. Nevertheless, future RCTs to evaluate the effects of polyphenols on the prevention and management of AMD progression should be developed. Efficient dosage, safety, and pharmacodynamics for treatment should also be further evaluated [44].

A systematic review conducted by Pameijer et al. found moderate certainty of evidence regarding the Mediterranean diet characterized by a high intake of fruit, vegetables, grains, nuts, fish, red wine, and olive oil and limited consumption of red meat. Thus, it may be a recommended approach to AMD management, and it is associated with a lower risk of AMD progression. Furthermore, the use of antioxidant supplements with this kind of diet were also associated with a reduced risk of progression [45].

## 4. Conclusions

The findings of this review suggest that Vit D and polyphenols may play important roles as nutritional modifiable protective factors associated with risk reduction of AMD progression, although more research with longer follow-up studies is important to better understand their roles in different stages of the disease. Vit D levels were lower in AMD patients compared with healthy subjects. The associations between the serum Vit D level and functional outcomes, such as contrast sensitivity, in AMD patients should be explored in future research. Dietary polyphenols have been shown to reduce progression in AMD's course and the expression of angiogenic factors VEGF in neovascular AMD. However, more research is necessary to understand their roles in AMD prevention and management.

**Author Contributions:** Conceptualization, N.F., M.C.A. and C.L.; methodology, C.L.; validation, N.F., M.C.A. and C.L.; formal analysis, N.F.; investigation, N.F.; resources, N.F.; data curation, N.F.; writing—original draft preparation, N.F.; writing—review and editing, C.L.; visualization, N.F.; supervision, M.C.A. All authors have read and agreed to the published version of the manuscript.

**Funding:** This research received no external funding.

**Informed Consent Statement:** Not applicable.

**Data Availability Statement:** Not applicable.

**Conflicts of Interest:** The authors declare no conflict of interest.

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
