# Peer review of "The Roles of Vitamin D and Polyphenols in the Management of Age-Related Macular Degeneration: A Narrative Review"

_futurepharmacol, doi:10.3390/futurepharmacol3010020_

Round 1

Reviewer 1 Report

1)    Regarding Lines 34-36, it is global incidence or specific to Europe. It will be beneficial to provide global numbers and then can specify the countries with highest incidences. 

2)    Provide references in lines 43 to 45 and lines 55 to 56. 

3)    In line 44, some nutritive supplement, explain or list some of the supplements. 

4)    Line 54, provide a few examples of chronic diseases that are associated with deficiency of vitamin D. 

5)    Line 55, no proper explanation given why women are more affected by Vit D deficiency.

6)    Lines 64 to 66 seems out of place. Please provide how anti-VEGF side effects are linked to AMD and flavonoids can be beneficial in that scenario. 

7)    Section 3.2, explain changes that occur during the progression of AMD such as damaged macula, drusen growth, or formation of new vessel. Also, label figure 1 in similar way so that reader can understand the changes you are trying to explain. 

8)    Reference(s) required in lines 123 to 124, 126-131.

9)    Explain what is mfERG ring 1 and 2.  

10) Majority of references are written as.[9], it should be [9]. Correction required throughout the manuscript. 

11) Lines 184 to 189, what happens to vitamin D level in different stages of AMD? 

12) Discussion is too long and a lot of it is reptation of the results section. You should cut it short and try to provide an overall picture of the literature; how these studies are flowing together. 

Author Response

Response to Reviewer 1

Point 1: Regarding Lines 34-36, it is global incidence or specific to Europe. It will be beneficial to provide global numbers and then can specify the countries with highest incidences.

R1. We added the global numbers worldwide and specified global incidence in Europe as suggested.

Lines 32-39: “Approximately 67 million people worldwide are currently affected by this disease, and it is expected an increase of its prevalence and incidence until 2050 in the European population [1]. AMD is responsible for 8.7% of blindness throughout the world and it is the main cause of visual impairment in developed countries in individuals aged 60 years or more [2]. The incidence of AMD is variable. The incidence of late AMD is higher Europe, while Africa has the highest incidence of early AMD. Annual incidence of late AMD in American whites was 0.35% and 0.14% in Europeans [3].”

Point 2: Provide references in lines 43 to 45 and lines 55 to 56.

R2. We have provided references in those lines as requested.

Point 3: In line 44, some nutritive supplement, explain or list some of the supplements.

R3. We added information on nutritive supplements and an explanation about their antioxidative properties.

Line:  52-57: “Some nutritive supplements with antioxidants properties are based on the Age-Related Eye Disease Study formulation (AREDS 1 and AREDS 2; e.g., Nutrof®, Remiren®, Meralut® and Vitol®). These supplements also contain vitamin D (Vit D) and polyphenols (blueberry and resveratrol) that contribute to improve retinal antioxidant defense. Vitamin D and dietary polyphenols provide a protective effect against the damage caused by free radicals [5].”

Point 4: Line 54, provide a few examples of chronic diseases that are associated with deficiency of vitamin D.

R4. We added the requested information.

Lines 65 to 66: “(…) the immune system, and its deficiency has been associated with differing chronic diseases such as auto immune and cardiovascular diseases, osteoporosis, diabetes, and cancer [7].”

Point 5: Line 55, no proper explanation given why women are more affected by Vit D deficiency.

R5. This information was taken from a Turkish study. Reasons may be related to environmental and social factors (women spend less time outdoors and type of clothing that does not allow the absorption of sunlight). As these results may be specific to the studied population, we have deleted this sentence.

Point 6: Lines 64 to 66 seems out of place. Please provide how anti-VEGF side effects are linked to AMD and flavonoids can be beneficial in that scenario.

R6. As requested, we have provided information on how anti-VEGF side effects are linked to AMD and flavonoids.

Lines 78-81: “Polyphenols have a preventive effect against chronic oxidative stress reducing anti-vascular endothelial growth factor (VEGF) side effects. Resveratrol and its analogs are promising agents that can improve the anti-VEGF therapy in exudative AMD [12].”

Point 7: Section 3.2, explain changes that occur during the progression of AMD such as damaged macula, drusen growth, or formation of new vessel. Also, label figure 1 in similar way so that reader can understand the changes you are trying to explain.

R7. As suggested, we have explained the changes that occur during the progression of AMD using the Age-Related Eye Disease Study (AREDS) classification categories. Additionally, we have revised the labelling of figure 1.

Line  128-135: “According to the AREDS classification, AMD can be divided in four categories based on ocular fundus examination and existence of drusen, atrophy, and neovascularization: Category 1 - no AMD if 0-5 small drusen are present; Category 2 – Early AMD if multiple small drusen or few intermediate-sized drusen are present, or macular pigmentary changes; Category 3 - intermediate AMD characterized by extensive intermediate-sized drusen, or at least one large drusen, or GA not-involving the macular center; and Category 4 - advanced AMD determined by GA involving the macular center or any sign of choroidal neovascularization (CNV) with vision loss [15].”

Point 8: Reference(s) required in lines 123 to 124, 126-131.

R8. We have provided references in those lines as requested.

Point 9: Explain what is mfERG ring 1 and 2.

R9. We have explained mfERG ring 1 and 2.

Lines  343-348: “The results of the trial showed increased average response in amplitude density in ring 1 (concentric annular central retinal region centered on the fovea limited area of 0º-2.5º) and ring 2 (concentric annular paracentral retinal region centered on the fovea limited area of 2.5º-5º) [38], suggesting a functional improvement of the pre-ganglionic retinal elements located in the 0–5 central retinal degrees [13].”

Point 10: Majority of references are written as.[9], it should be [9]. Correction required throughout the manuscript.

R10. As suggested, we have corrected the references throughout the manuscript.

Point 11: Lines 184 to 189, what happens to vitamin D level in different stages of AMD?

R11. None of the studies was able to explain in detail what happens to vitamin D level in different stages of AMD. We have provided this information in lenes 180 to 182.

Lines 180-182: “Nevertheless, the authors concluded that it is difficult to determine the precise correlation between vit D deficiency and AMD progression and emphasize the different 25(OH)D concentration blood levels used by the different studies [21].

Point 12: Discussion is too long and a lot of it is reptation of the results section. You should cut it short and try to provide an overall picture of the literature; how these studies are flowing together.

R12. The discussion section was reformulated as requested.

Reviewer 2 Report

Authors have introduced as in narrative review the effects of vit D and flavonoids on the AMD.

Why there is investigated vit D and flavonoids in the same manuscript? Is there some connection with those?

Vit D is introduced quite comprehensivlye but flavonoids still needs more effort. As well need to know why those two group are included into this manuscript and not done as separately.

Reference list is quite short. Especially flavonoids probably is more as single spesific flavonoids?

Abstract

- Please check if correct ”… protective effect of vitamin D and AMD.

- Please, tell also how? In or after next sentence ” Vitamin D and flavonoids may play an important role as nutritional modifiable protective factors to reduce the risk of AMD progression.

Introduction

- Add wavelengths to the risks UV and blue light.

- add as well few example source of flavonoids from diet.

Material and Methods

- please check all abbreviations that open when mentioned first time e.g. RCT

- Please, explain this ” Studies included in previous reviews/meta-analysis were excluded from this study.” Why those were excluded?

Results

- Figure 1. Please indentify three different stages shown in pictures example by A to C and refer in figure legend. There should mention someway differences between three pictures shown in figure 1.

- Why malabsorption is refered under title of D-vit?

- Give in the text some conclusion related to references in table 1 and reason why results may differ from each other. As well for table 2, give some conclusion related to the reference researches.

- check line 166

- modify reference-style related to webpage.

- There is mentioned that there is only few studies related to flavonoids. This need to open more or explain. Could there be more research done example with different single flavonoids?

Conclusion need to justify shortly based to the observations of the review

- Next sentence need to justify more ” Vit D and flavonoids may be important biomarkers in the management of AMD,” as well inflammation is mentioned maybe first time in conclusion? What it means and how connected to other parts of the review? At least need to open little bit at this point.

Author Response

Response to Reviewer 2

Authors have introduced as in narrative review the effects of vit D and flavonoids on the AMD.

  1. We thank reviewer 2 for the helpful and insightful comments. We have considered all the comments carefully and thoughtfully in this version on the manuscript.

Point 1: Why there is investigated vit D and flavonoids in the same manuscript? Is there some connection with those?

R.1. The terms “flavonoids” was replaced by “polyphenols” that included flavonoids (anthocyanins and quercetin) and other polyphenols (resveratrol and curcumin). Vitamin D and polyphenols were included in the same manuscript because both have antioxidative and anti-inflammatory properties which have an impact on oxidative stress and inflammation processes linked with AMD pathogenesis. This information was added in lines 86-91.

Point 2: Vit D is introduced quite comprehensively but flavonoids still need more effort. As well need to know why those two group are included into this manuscript and not done as separately.

R.2. Please see answer R1. Additionally, we have described polyphenols in better detailed.

Point 3: Reference list is quite short. Especially flavonoids probably is more as single specific flavonoids?

R.3. We have revised and increased the list of references. Additionally, we replaced flavonoids by Polyphenols. Anthocyanins, quercetin, and resveratrol were the selected polyphenols.

Lines 201-208 “Polyphenols or phenolic compounds are a group of more than 8000 diverse phytochemicals with a phenolic structure. They are found in plants and beverages, such as vegetables, fruits, chocolate, tea and wine. Polyphenols are classified according to their chemical structure and can be divided into groups: flavonoids (flavanols, flavan-3-ols, flavones, flavanones, and anthocyanins), phenolic acids, phenolic amides, and other polyphenols (curcumin, resveratrol). Previous research demonstrated that dietary polyphenols have a beneficial role in the prevention of chronic diseases and may improve human health through their antioxidant and anti-inflammatory activities [22]”

Point 4: Abstract

- Please check if correct ”… protective effect of vitamin D and AMD.”

R4.1. We have corrected this typo. The word “and” was replaced by “in AMD progression”.

- Please, tell also how? In or after next sentence” Vitamin D and flavonoids may play an important role as nutritional modifiable protective factors to reduce the risk of AMD progression.”

R4.2. Through dietary intake of food sources rich in this micronutrient or through supplementation with the correct doses, considering the health status and patient´s diet. This information was added on lines 280 to 283; lines 359 to 362: “Vit D may protect the retinal pigmentary epithelium (RPE) and choroidal cell function, layers that are directly related with AMD development. Thus, increasing 25(OH)D levels through dietary intake such as vit D-enriched foods and supplementation may be beneficial [7].”; “There is limited research on promising phytochemicals with antioxidant and anti-inflammatory properties. Angiogenic factors VEGF and hypoxia-inducible factor 1α (HIF-1α) expression could be reduced by polyphenols. Flavonoids appear to play an important role on angiogenesis inhibitors, but this effect is unclear [11].”

Point 5: Introduction

- Add wavelengths to the risks UV and blue light.

R.5.1. We added this information in line 42-43; 331 to 333: “(…) exposure to ultraviolet [UV; 240-300] and blue light [415-455]); It is important to note, that reduced ultraviolet B (UVB) (290-315 nm) at relatively high latitudes, particularly during the winter, may decrease cutaneous absorption and synthesis of vit D3.”

- add as well few example source of flavonoids from diet.

R.5.2. Flavonoids were substituted for phenolic groups. We have added examples, as suggested. Lines 201-203: “Polyphenols or phenolic compounds are a group of more than 8000 several and diverse phytochemicals with a phenolic structure, found in plants and beverages, such as vegetables, fruits, chocolate, tea and wine.”

Point 6: Material and Methods

- please check all abbreviations that open when mentioned first time e.g. RCT

R.6.1. We have verified all abbreviations as suggested.

- Please, explain this ”Studies included in previous reviews/meta-analysis were excluded from this study.” Why those were excluded?

R.6.2. Observational studies and RCTs that have already been included and analyzed in previous meta-analyzes were excluded from our results table, but the overall results from meta-analysis have been included in the discussion of our narrative review.

Point 7: Results

- Figure 1. Please identify three different stages shown in pictures example by A to C and refer in figure legend. There should mention someway differences between three pictures shown in figure 1.

R.7.1. As suggested, we have revised the figure legend. We also added SD-OCT images add revised the classification emphasizing the three different AMD stages.

Lines 121-126: “Figure 1. Ocular fundus retinography (A) and spectral domain – optical coherence tomography (B) showing the progression from intermediate to advanced AMD; from left to right: (A.1 B.1) Large, confluent, soft drusen in high risk intermediate AMD, retinal pigment epithelium (RPE) migration and pericentric geographic atrophy; (A.2 B.2) Occult choroidal neovascularization, pigmentary clumping and RPE serous detachment with tear in advanced stage; (A.3 B.3) End stage AMD with disciform scar (Araujo, M. e Fernandes, N. courtesy).”

- Why malabsorption is refered under title of D-vit?

R.7.2. We have revised the text. Lines: 150-153: “Some health conditions cause malabsorption vitamin D such as cystic fibrosis, inflammatory bowel diseases, gastric bypass surgery, and intestinal lymphangiectasia. Therefore, the risk for vitamin D deficiency increases in those individuals [18].”

- Give in the text some conclusion related to references in table 1 and reason why results may differ from each other. As well for table 2, give some conclusion related to the reference researches.

R.7.3. We have revised table 1 and 2 as well as the conclusions as suggested.

- check line 166

R.7.4. We have revised this line.

- modify reference-style related to webpage.

R.7.5. We have revised the reference-style related to the webpage link.

- There is mentioned that there is only few studies related to flavonoids. This need to open more or explain. Could there be more research done example with different single flavonoids?

R.7.6.: We have discussed the role of polyphenols in more detail. We also added information on clinical trials that are still recruiting patients. Additionally, we added suggestions for further research.

Point 8: Conclusion need to justify shortly based to the observations of the review

R.: We have revised the conclusion.

- Next sentence need to justify more ” Vit D and flavonoids may be important biomarkers in the management of AMD,” as well inflammation is mentioned maybe first time in conclusion? What it means and how connected to other parts of the review? At least need to open little bit at this point.

R.8.1: This sentence was revised. Lines 393-395: “Dietary Polyphenols have been shown to reduce progression in AMD course and the expression of angiogenic factors VEGF in neovascular AMD. Although, more research is necessary to conclude about their role in eye health nutrition.”

Reviewer 3 Report

The manuscript proposed a narrative review of the literature on the role of vitamin D (particularly through its anti-inflammatory and antioxidant effect) and flavonoids in the prevention of AMD and the possible improvement of related clinical conditions. The authors selected both interventional and observational studies for this purpose. From the selected articles, fragmented and mixed data are highlighted with a possible implication of vitamin D in the preventive phase, which emerged above all in observational studies.

Even if the subject is very interesting, the impression is of an incomplete deal with the subject and sometimes a too concise description of some selected works. More in detail:

- Prevention and treatment should be the main aim of the review and therefore the two aspects should be organized more organically in the text. Instead, it appears that the authors deal with the two aspects without pooling the data for each. It could also help with a table/image showing the current evidence of efficacy in case of prevention or treatment, in case of sufficient or insufficient circulating levels and the effect of intervention with physiological or pharmacological dosage.

- The selection is not clear, especially the fact that the authors claim not to have discussed studies already treated by other meta-analyses, systematic reviews and more generally narrative reviews. I recommend selecting the sources, excluding only those already available in the meta-analyses. During the discussion, the authors will be able to briefly describe the results of the meta-analyses and start from these results to integrate with the other selected works. This will hint at the background (with high evidence data such as meta-analyses) and update the evidence with the remaining articles.

- The manuscript could benefit from focusing only on vitamin D. About flavonoids, the authors claim to have found few articles but describe only 1.

- Although this is not a systematic review, a systematic method has been described and therefore it may be useful to include a flow chart describing the source selection process.

- (OPTIONAL) if the authors are able, a qualitative evaluation of the works would enrich the manuscript

- A brief description of antioxidant and anti-inflammatory molecular mechanisms would fit a narrative review

- Among the exclusion criteria, it would be necessary to speak about preclinical studies while among those for inclusion, even if they had not emerged, the authors should also include retrospective as well as prospective and cross-sectional studies.

- In lines 121-123, the authors forgot to include foods and fortified foods among the sources. In northern Europe, for example, these represent the main source for circulating levels of vitamin D. However, a reference is needed for the authors' statement.

- The paragraph on page 6 should have a specific title and the description of the work of Merle et al. should be moved to the previous paragraph

- The sentence on line 166 should be revised

- The works' descriptions on lines 180-189 are too hasty and superficial

- I recommend removing the paragraph on flavonoids (and revising the title and discussion accordingly)

Author Response

Reply review report 3

The manuscript proposed a narrative review of the literature on the role of vitamin D (particularly through its anti-inflammatory and antioxidant effect) and flavonoids in the prevention of AMD and the possible improvement of related clinical conditions. The authors selected both interventional and observational studies for this purpose. From the selected articles, fragmented and mixed data are highlighted with a possible implication of vitamin D in the preventive phase, which emerged above all in observational studies.

Even if the subject is very interesting, the impression is of an incomplete deal with the subject and sometimes a too concise description of some selected works. More in detail:

  1. We thank reviewer 3 for the helpful and insightful comments. We have considered all the comments carefully and thoughtfully in this version on the manuscript.

Point 1: Prevention and treatment should be the main aim of the review and therefore the two aspects should be organized more organically in the text. Instead, it appears that the authors deal with the two aspects without pooling the data for each. It could also help with a table/image showing the current evidence of efficacy in case of prevention or treatment, in case of sufficient or insufficient circulating levels and the effect of intervention with physiological or pharmacological dosage.

R.1. The paper was re-organized highlighting the AMD progression aspect. Tables 1 show association between vit D levels and AMD and table 2 the effect of vit D and polyphenols dietary intake and supplementation dosages, and the effects in retina and choroid function-structure.

Point 2: The selection is not clear, especially the fact that the authors claim not to have discussed studies already treated by other meta-analyses, systematic reviews and more generally narrative reviews. I recommend selecting the sources, excluding only those already available in the meta analyses. During the discussion, the authors will be able to briefly describe the results of the meta-analyses and start from these results to integrate with the other selected works. This will hint at the background (with high evidence data such as meta-analyses) and update the evidence with the remaining articles.

R.2. We have excluded studies already available in previously meta-analyses. In the discussion section we have described the results of meta-analysis and updated the evidence with the remaining articles.

Point 3: The manuscript could benefit from focusing only on vitamin D. About flavonoids, the authors claim to have found few articles but describe only 1.

R.3. We have revised the term “flavonoids” to “polyphenols” giving particular attention to flavonoids (anthocyanins and quercetin), but also other polyphenols such as resveratrol and curcumin. We have revised the title of the manuscript and updated the number of included studies.

Point 4: Although this is not a systematic review, a systematic method has been described and therefore it may be useful to include a flow chart describing the source selection process.

R.4. Reviewer number 4 asked to delete the methods section and to organize the manuscript as traditional narrative review. Thus, the methods section was deleted, and a flow chart would not fit in the new organization of the manuscript.

Point 5: (OPTIONAL) if the authors are able, a qualitative evaluation of the works would enrich the manuscript

R.5. We agree and appreciate your comment. However, reviewer 4 asked for a traditional narrative review and a qualitative evaluation would fit better a systematic review. Thus, we have not included a qualitative evaluation of the included manuscripts.

Point 6: A brief description of antioxidant and anti-inflammatory molecular mechanisms would fit a narrative review.

  1. 6. We have included a description of antioxidant and anti-inflammatory mechanisms of vitamin D and polyphenols. Additionally, we have included a figure with the molecular structures.

Point 7: Among the exclusion criteria, it would be necessary to speak about preclinical studies while among those for inclusion, even if they had not emerged, the authors should also include retrospective as well as prospective and cross-sectional studies.

R.7. Preclinical studies were excluded from our research. We have added this information to the manuscript. Retrospective as well as prospective and cross-sectional studies were included if available.

Line 94 to 95: “We excluded preclinical studies and studies included in previous meta-analysis.”

Point 8: In lines 121-123, the authors forgot to include foods and fortified foods among the sources. In northern Europe, for example, these represent the main source for circulating levels of vitamin D. However, a reference is needed for the authors' statement.

R.8. In the revised version, we added information on foods fortified foods.

Lines 145-148: “Deficiency in vit D levels is common in older aged people. Although vit D may be taken in the form of nutritional supplements or vit D food fortification (e.g. cow’s milk, margarine, orange, plant-based milk, cereals, and bread) [17], vit D is primarily produced in the skin, and its production may be impaired in the elderly due to skin atrophy [10].”

Point 9: The paragraph on page 6 should have a specific title and the description of the work of Merle et al. should be moved to the previous paragraph

R.9. The paper was re-organized in new sections.

Point 10: The sentence on line 166 should be revised

R.10. The sentence was revised as requested.

Line 162: In Icel et al. case-control study (Table 1), macular microvasculature in”…

Point 11: The works' descriptions on lines 180-189 are too hasty and superficial

R.11. We have revised the descriptions on those lines.

Lines 175-182: “In their study, plasma 25(OH)D concentrations were defined in deficiency categories as severe (<10 ng/mL), deficiency (10-19 ng/mL), insufficiency (20-29 ng/mL) and sufficiency (≥30) [21]. The study results revealed that vit D levels were decreased in patients affected by AMD compared to controls. Nevertheless, the authors concluded that it is difficult to determine the precise correlation between vit D deficiency and AMD progression and emphasize the different 25(OH)D concentration blood levels used by the different studies [21].”

Point 12: I recommend removing the paragraph on flavonoids (and revising the title and discussion accordingly)

R.12. After a new literature search, as well as a review of the existing literature in the manuscript, we replaced flavonoids by polyphenols as this is a more comprehensive group (flavonoids group: anthocyanins and quercetin; other polyphenols: resveratrol and curcumin). We included vitamin D and polyphenols in the same manuscript because both have antioxidative and anti-inflammatory properties which have an impact on oxidative stress and inflammation processes linked to AMD pathogenesis.

Reviewer 4 Report

In this work, the authors narrowly review the vitamin D and flavonoids in preventing and managing Age-related Macular Degeneration. However, the whole discussion and summary are too shallow, even it is a narrow review. Also, the logic and format are also not like a review paper.  And no need to fully discuss the reference sources in detail, especially in a good review paper. The contents, tables, and figures, not only in quality but also the content richness, do not meet the journal’s requirements. Hence, I recommend this paper should have a to enrich more content, the author summarizes the field, as well as personal opinions, and the future development direction, and then resubmits to the journal.  Specifical comments are listed below. 

1. The abbreviation in the abstract should also be given.

2. The logic of the whole paper should be adjusted, as a review paper, not need to give a separate method section in detail. 

3. What is the current development of the diagnosis of the disease or vitamin D? It also should be mentioned. 

4. English still need to be polished. 

Author Response

Response to Reviewer 4

Comments: In this work, the authors narrowly review the vitamin D and flavonoids in preventing and managing Age-related Macular Degeneration. However, the whole discussion and summary are too shallow, even it is a narrow review.

R: We thank reviewer 4 for the helpful and insightful comments. We have considered all the comments carefully and thoughtfully in this version of the manuscript. We have revised the whole discussion and summary.

Also, the logic and format are also not like a review paper.

We have revised the logic and format of the manuscript.

And no need to fully discuss the reference sources in detail, especially in a good review paper.

We have discussed the reference sources in more detail in this revised version.

The contents, tables, and figures, not only in quality but also the content richness, do not meet the journal’s requirements. Hence, I recommend this paper should have a to enrich more content, the author summarizes the field, as well as personal opinions, and the future development direction, and then resubmits to the journal.Specifical comments are listed below.

Contents, tables, and figures were revised accordingly.

Point 1: The abbreviation in the abstract should also be given.

R1: We have revised all the abbreviations.

Point 2: The logic of the whole paper should be adjusted, as a review paper, not need to give a separate method section in detail.

R2. We have revised the logic and format of the manuscript. The method section was removed.

Point 3: What is the current development of the diagnosis of the disease or vitamin D? It also should be mentioned.

R3. We have included a description of the current development of the diagnosis of the disease or vitamin D as suggested. The sections of the paper were revised to a new section on clinical diagnosis and other two for vitamin D levels and effects of supplementation.

Point 4: English still need to be polished.

R4. We have revised the English.

Round 2

Reviewer 2 Report

- start from line 32. Please add still the estimated prevalence until 2050.

- line 216. ” test with a progression from 25% to 60% from day 60 to day 90,

is this line with previous text in the sentence?  At least explain what this mean? Better situation/vision from 25% to 60% or worsen from 25 to 60.

- in discussion there is now lot of info about D-vit. There is not much anything about polyphenols except curcumin. It is little bit confusing highlight curcumin. There should add some general infor about polyphenols related to AMD and then introduce as well some other flavonoids/polyphenols examided related to AMD.

- In conclusion, it is little bit confusing ” to conclude about their role in eye health nutrition” Pleas explain or modify.

- if there is only some selected polyphenols introduced, it should mention that group is huge and in this review there is concentrated into few of them and why.

-Still there could be more references example in discussion related to polyphenols. It was mentioned but changes related to that were small.

Author Response

Response to Reviewer 2

We appreciate the opportunity to revise our paper.

1 - start from line 32. Please add still the estimated prevalence until 2050. 

R1: We added the estimated prevalence as requested.  

Line 33 - and it is expected an increase “by 15%” of its prevalence.

2 - line 216. ” test with a progression from 25% to 60% from day 60 to day 90,”

is this line with previous text in the sentence?  At least explain what this mean? Better situation/vision from 25% to 60% or worsen from 25 to 60.

R2: We have revised the sentence.  

Line 217 – “(…) and reduced distortion in Amsler’s Grid test with an improvement in central vision from 25% to 60%, since day 60 to day 90, respectively, after the treatment.”

3 - in discussion there is now lot of info about D-vit. There is not much anything about polyphenols except curcumin. It is little bit confusing highlight curcumin. There should add some general infor about polyphenols related to AMD and then introduce as well some other flavonoids/polyphenols examided related to AMD.

R3: We have revised the discussion section as suggested.

Lines 361-367 – “There are more than 8000 diverse phytochemicals polyphenols. However, only a few have been studied in RCTs and observational studies that included patients with AMD. AMD is an age-related disease associated with chronic oxidative stress. Thus, antioxidative, anti-inflammatory and antiaging properties of polyphenols may play an important role in the prevention and management of AMD [31]. Natural antioxidants may act by slowing the progression of AMD, contributing to prevention of irreversible vision loss [39]. Flavonoids are the predominant dietary polyphenols. Flavonoids act by decreasing oxidative stress and inflammatory processes, as well as producing angiogenesis inhibitors. A regular diet rich in these compounds can have a relevant role on the prevention of AMD, as well as inhibiting progression of AMD [40]. Epigallocatechin gallate found in green tea, inhibits ROS, angiogenesis, VEGF, apoptosis of retinal ganglion cells and protects against mitochondrial damage. While quercetin found in fruits and vegetables inhibits ROS, VEGF, pro-inflammatory molecules, and apoptosis of the neurons, protecting RPE cells in age-related diseases [41]. Quercetin may also play an important role in AMD progression due to its capacity to reduce oxidative damage that induces inflammation and to reverse photoreceptor apoptosis and retinal degeneration [39].”

Lines 381-393 – “Other groups of polyphenols may also have an important role in AMD progression. Curcumin (Curcuma longa) is a polyphenolic compound with antioxidant property which could be combined with another phenolic, resveratrol. Curcumin can inhibit lipid peroxidation, ROS, and VEGF. It also reduces pro-inflammatory cytokines and DNA damage through increasing antioxidant enzymes. Resveratrol contributes to inhibit oxidative stress, VEGF, lipid peroxidation, decreases inflammatory processes and increases glutathione [41]. A retrospective case-control study conducted by Allegrini et al. demonstrated that a curcumin supplement with AREDS2 compounds, astaxanthin and resveratrol, combined with anti-VEGF injection improved functional outcomes with a reduction of the injection therapy [42]. Nevertheless, additional evidence is necessary to evaluate safety and efficacy of curcumin in AMD [39]. The therapeutic potential of resveratrol can be related with a reduction of anti-VEGF injection side effects. Future research to investigate the synergetic effects of resveratrol and anti-VEGF in neovascular AMD may be relevant to reduce the complications of this treatment [39].”

Lines 402-404 - “Anthocyanins are present in blueberries, raspberries, blackberries, strawberries, or red grapes. This flavonoid reduces the inflammation process in chronic conditions, namely Cyanidin-3-glucoside (C3G) that it is important in the prevention of retinal diseases [40]. “

Lines 410-431 – “Saffron (Crocus sativus) is a phytochemical with 150 compounds, with carotenoids, crocin and crocetin as the most organically active elements. Open-label longitudinal studies and double-blind RCTs demonstrated that saffron reduces progression of dry, mild to moderate and advanced stages of AMD [39]. Long-term consumption of polyphenolic compounds may protect against some chronic disorders such as cancers, cardiovascular and neurodegenerative diseases [44]. The efficacy of polyphenols, bioavailability, absorptive food and drug interactions, short- and long-term health effects and possibly health-promoting mechanisms should be investigated in future research [44]. With aging, the composition of the microbiome changes and several degenerative diseases are associated with microbiome aging. AMD is a multifactorial disease related with age, environmental, epigenetic, and genetic risk factors. Diet seems to have a deep impact on the onset and progression of AMD through anti-oxidative and anti-inflammatory mechanisms [45]. There is potential interaction between the immune system modulation, inflammation and diet on gut microbiota. Thus, gut microbes and their metabolites may play a key role in AMD [46]. Polyphenols, such as blueberry and green tea extract drink, have potential interactions with gut microbiota, as its metabolites may promote beneficial gut bacteria and inhibiting intrusive species [44]. Polyphenols seem to be one of the most important compounds inhibiting effects on AMD progression, through dietary intake of fruits and vegetables. Nevertheless, future RCTs to evaluate the effect of polyphenols in prevention and management of AMD progression should be developed. Efficient dosage, safety, and pharmacodynamics for treatment should also be further evaluated [47].”

4 - In conclusion, it is little bit confusing ” to conclude about their role in eye health nutrition” Pleas explain or modify.

R4: We have modified the sentence.

Line 448 – “Although, more research is necessary to understand their role on AMD prevention and management.”

5 - if there is only some selected polyphenols introduced, it should mention that group is huge and in this review there is concentrated into few of them and why.

R5: We have added this information as requested. Lines 361-362 – “There are more than 8000 diverse phytochemicals polyphenols. However, only a few have been studied in RCTs and observational studies that included patients with AMD.”

6 -Still there could be more references example in discussion related to polyphenols. It was mentioned but changes related to that were small.

R6: We added more references on polyphenols. However, we would like to highlight that the number of studies on polyphenols is much less than the studies on vitamin D.

Reviewer 3 Report

The authors improved their manuscript according to the Reviewers' requests. Thank you for your work

Author Response

Many thanks for the opportunity to revise our paper.

Reviewer 4 Report

The author made good revison, no comment anymore.

Author Response

(The authors gave the same response as above.)
